behaviour, ecology

enemy avoidance, infochemicals, hazard cues, phloem-feeding insects, cuticular hydrocarbons

**Author for correspondence:**
Alejandro Tena
e-mail: atena@ivia.es

# Parasitic wasps avoid ant-protected hemipteran hosts via the detection of ant cuticular hydrocarbons

Angelos Mouratidis[1,2], Sandra Vacas[3], Julieta Herrero[1], Vicente Navarro-Llopis[3], Marcel Dicke[2] and Alejandro Tena[1]

[1]Instituto Valenciano de Investigaciones Agrarias, Plant Protection and Biotechnology Research Center, Moncada, Spain
[2]Wageningen University, Laboratory of Entomology, Wageningen, The Netherlands
[3]Centro de Ecología Química Agrícola, Instituto Agroforestal del Mediterráneo, Universitat Politècnica de València, València, Spain

AM, 0000-0001-6358-4574; SV, 0000-0001-6911-1647; VN-L, 0000-0003-3030-3304; MD, 0000-0001-8565-8896; AT, 0000-0002-5001-4334

One of the most studied and best-known mutualistic relationships between insects is that between ants and phloem-feeding insects. Ants feed on honeydew excreted by phloem-feeding insects and, in exchange, attack the phloem feeders' natural enemies, including parasitic wasps. However, parasitic wasps are under selection to exploit information on hazards and avoid them. Here, we tested whether parasitic wasps detect the previous presence of ants attending colonies of phloem feeders. Behavioural assays demonstrate that wasps left colonies previously attended by ants more frequently than control colonies. This behaviour has a potential cost for the parasitic wasp as females inserted their ovipositor in fewer hosts per colony. In a further bioassay, wasps spent less time on papers impregnated with extracts of the ant cues than on control papers. Gas chromatography coupled with mass spectrometry analyses demonstrated that ants left a blend of cuticular hydrocarbons when they attended colonies of phloem feeders. These cuticular hydrocarbons are deposited passively when ants search for food. Overall, these results suggest, for the first time, that parasitic wasps of honeydew producers detect the previous presence of mutualistic ants through contact infochemicals. We anticipate such interactions to be widespread and to have implications in numerous ecosystems, as phloem feeders are usually tended by ants.

## 1. Introduction

One of the most studied and best-known mutualistic relationships among insects is the one formed between ants and hemipterans [1,2]. Ants feed on the sugary excretion of hemipterans, called honeydew. In exchange, ants improve the hygiene of honeydew-producing colonies by removing dead individuals and exuviae and by transporting honeydew producers to suitable parts of the plant. Moreover, ants protect and guard honeydew producers against their natural enemies, mostly predators and parasitic wasps [2]. Numerous examples demonstrate that this mutualistic relationship increases the population of the honeydew producer because ants reduce predation and parasitism [3–10]. When ants attend and defend the colonies of honeydew producers, they attack and even kill natural enemies of the honeydew producers. Parasitic wasps can be vulnerable to these attacks because they remain defenceless when they parasitize honeydew producers or feed on host haemolymph, i.e. host feeding [3,11–13].

However, parasitic wasps, just as other animals, are under selection to exploit information about their environment and dangers [14]. They can use

their sensory system to detect the presence of enemies via mechanical, visual and chemical cues. Once detected, parasitic wasps may adjust their behaviour accordingly [14,15]. For example, the aphid parasitoid *Aphidius ervi* Haliday (Hymenoptera: Braconidae) avoids patches previously exposed to its intraguild predator *Coccinella septempunctata* L. (Coleoptera: Coccinellidae) and, when it accepts exposed patches, it parasitizes fewer hosts [16]. This wasp responds to cuticular hydrocarbons that *C. septempunctata* deposits in its trails [17]. The detection of danger has obvious advantages, especially for species that are under strong selective pressure from organisms at higher trophic levels [14]. This is likely the case for parasitic wasps of honeydew-producing species that are tended by ants.

Apart from the visual and mechanical cues, parasitic wasps of honeydew-producing species may exploit the wide range of pheromones that ants use to communicate [1,18]. It has been demonstrated that insects from several orders and with very different biologies can detect areas marked with ant cues [19–22]. We hypothesize that parasitic wasps of honeydew-producing species may also modify their behaviour when they encounter colonies previously attended by ants. Here, we first: (i) determined whether parasitic wasps that parasitize honeydew producers can detect the previous presence of mutualistic ants in the colony of the honeydew producers. For this, we measured changes in colony detection, time spent foraging in the colony, as well as the proportion and number of times that parasitic wasps left the colonies when they searched in colonies non-exposed and previously exposed to ants. We expect that if parasitic wasps detect the previous presence of ants in the colony using ant cues, they will locate colonies equally but will spend less time in colonies previously exposed to ants and leave these colonies more frequently. Then, we (ii) evaluated the number of hosts stung per colony. We expect that if parasitic wasps detect the previous presence of ants in the colony, they will sting more hosts in non-exposed colonies than in colonies exposed to ants. Finally, we (iii) identified the chemical cues involved. We have addressed these aims through behavioural and chemical assays using laboratory (queenless) and field (queenright) ant nests.

## (a) The study system

The citrus mealybug *Planococcus citri* (Risso) (Hemiptera: Pseudococcidae) is a cosmopolitan phytophagous insect that is distributed in more than 87 countries of tropical and subtropical regions of the world, frequently cited in fruit orchards, such as citrus and grapevine [23–25], as well as in greenhouse horticulture [26]. *Planococcus citri* excretes large amounts of honeydew, and its colonies are usually tended by ants [10,27]. In the Mediterranean Basin, one of the main ant species that forms a mutualistic relationship with *P. citri* is the native ant *Lasius grandis* (Forel) (Hymenoptera: Formicidae) [4,27,28]. The main parasitic wasp of *P. citri* is the native encyrtid parasitic wasp *Anagyrus vladimiri* Triapitsyn (formerly known as *Anagyrus* sp. near *pseudococci* (Girault)) (Hymenoptera: Encyrtidae) [29]. This synovigenic and solitary endoparasitoid is attacked by ants attending mealybugs [7,9]. For example, the Argentine ant *Linepithema humile* (Mayr) (Hymenoptera: Formicidae) reduces the parasitism rate of *A. vladimiri* when it attends mealybug colonies [7,9].

# 2. Material and methods

## (a) Insects: mealybugs and parasitic wasps

*Planococcus citri* was obtained from the State Insectary of Valencia (Spain) where they were reared on butternut squash (*Cucurbita moschata* L.) according to methods described by Daane *et al.* [30]. Subsequently, they were reared in the facilities of Instituto Valenciano de Investigaciones Agrarias (IVIA) on green bean pods (*Phaseolus vulgaris* L.) for the experiment with mealybug colonies exposed to field ant nests and on potato sprouts for the experiment with mealybug colonies exposed to queenless ant nests. Mealybugs were kept in plastic boxes (30.5 × 24.5 × 20 cm) with a hole (approx. 15 × 15 cm) covered with muslin on top under laboratory conditions (23 ± 3°C).

Pupae of *A. vladimiri* were obtained from Koppert Biological Systems S.L. (Águilas, Murcia, Spain). At their arrival, pupae were introduced into wood-and-glass rearing boxes (51 cm × 51 cm × 41 cm) with the side walls covered with fine mesh for ventilation. For the two first bioassays (colony use), drops of honey were provided directly on the walls as food for the emerging adults and were replenished daily. The rearing boxes were kept in a climatic cabinet (Sanyo MLR-251) at 25 ± 1°C, 70 ± 10% RH and 16 L : 8 D photoperiod. For both assays, freshly emerged wasps of both sexes were collected daily and transferred to another rearing box with bean pods or potato sprouts infested with *P. citri* in order to mate and gain oviposition experience for 24 h. Then, females were individualized in 3.0 cm × 0.8 cm diameter glass vials with a drop of honey on the wall, sealed with cotton wool and kept in the climatic cabinet.

For the last bioassay (paper impregnated with ant cues), *A. vladimiri* pupae were introduced in the same boxes and climatic cabinet than in the previous assays, but without honey to obtain starved females. Freshly emerged females were collected daily and individualized in 3.0 cm × 0.8 cm diameter glass vials, sealed with wet cotton wool and kept in the climatic cabinet. Females were between 24 and 48 h old when they were used in the assay.

## (b) Mealybug colonies exposed to queenless ant nests

To obtain the ant nests, eight nest fragments were collected in a citrus orchard at Instituto Valenciano de Investigaciones Agrarias (Moncada, Spain) by digging up the nest from the ground. In the laboratory, the experimental nests were placed in eight plastic boxes (30.5 × 24.5 × 20 cm). The inner walls of the boxes were lined with Fluon® at 60% (Aldrich Chemistry) to prevent ants from escaping. All nest fragments were queenless and composed of approximately 200 workers. Nest fragments were kept in the laboratory (23 ± 3°C) and provided with a solution of water, honey and yeast at 1 : 4 : 1 on a piece of aluminium foil twice a week. Water was provided in a glass vial (15 × 1.5 cm in diameter) tapped with a piece of cotton wool in the middle of the vial. The vial was covered with a piece of aluminium foil and ants used it as a nest. Forty-eight hours prior to assays, food was removed to starve the ants and homogenize their feeding status.

To obtain mealybug colonies of similar size, potato sprouts were infested with 20–25 second instar to pre-ovipositional adult mealybugs, as *A. vladimiri* shows a host preference for older mealybugs [32]. Potato sprouts were individually placed into plastic boxes (16.5 × 11 × 6 high cm) with walls coated with Fluon® both inside and outside the boxes. Sprouts were maintained with centrifuge tubes filled with bacteriological agar (20 g l$^{-1}$), with the sprout base inserted directly into the agar. Tubes were sealed with Parafilm® to avoid ants digging. After infestation, mealybugs could settle and feed for 48 h.

To obtain mealybug colonies that had been in contact with ants (ant-exposed colonies), plastic boxes with the infested potato sprouts were connected to ant nests. Starved ants were allowed

to forage in a mealybug colony by temporarily connecting the colony with the plastic box with a wire as a bridge (electronic supplementary material, figure S1A). Twenty-four hours later, the wire was carefully removed while ants were not using it. Immediately, potato sprouts were moved to an experimental arena to observe the behaviour of the female wasp. Arenas consisted of a polystyrene plastic box ($10 \times 14 \times 14$ cm) with a lateral hole ($4 \times 9$ cm) covered with muslin. Inside the arena, the centrifuge tube with the infested potato sprout was placed vertically on a silicon base to elevate the colony and improve the accuracy of the observations. The same procedure was followed with mealybug colonies that had not been exposed to ants. The non-exposed and the ant-exposed treatments were replicated 56 and 56 times, respectively.

## (c) Mealybug colonies exposed to field queenright ant nests

To corroborate the results obtained with queenless ant nests, we carried out a similar experiment but using field queenright ant nests. This is because the presence of queens in ant nests affects the behaviour of workers in some ant species and, therefore, might change the chemical cues left by workers [33].

To obtain mealybug colonies of similar size, green bean pods were used as the plant substrate for the mealybugs. Prior to inoculation, one side of the bean pods was submerged in red paraffin wax to minimize the area for the mealybugs to settle and facilitate the observations (based on the methodology described by Cebolla *et al.* [34]). Bean pods were infested with 20–25 second instar to pre-ovipositional adult *P. citri*. Mealybugs were allowed to settle and feed for 24 h.

To obtain mealybug colonies that had been in contact with field ants, 12 citrus trees with high *L. grandis* activity were selected in a 15-year-old IVIA organic citrus orchard [*Citrus sinensis* (L.) Osbeck (Var. Navelate)]. A plastic box was used to expose the beans infested with mealybugs to foraging ants. Boxes were $38.5 \times 32 \times 25$ cm and had four small holes (0.5 cm diameter) in one side to allow the entry of ants, and two big lateral holes ($15 \times 10$ cm) covered with mesh for ventilation. Boxes were placed near the base of a citrus tree's trunk, and the four holes of the boxes were placed next to the path of *L. grandis* that were foraging in the tree canopy (electronic supplementary material, figure S1B). In each box, infested beans were kept with insect pins on flower sponge fixed with silicone to the base of the box. Ants had access to the infested beans for 24 h. Then, the box was carefully removed from the trunk, while ants were not present. Immediately after collection, infested beans were transported back to the laboratory in sterile boxes using disposable nitrile gloves. The same procedure was followed to obtain mealybug colonies non-exposed to ants, but the holes were blocked to exclude ant attendance. Experimental arenas for the behavioural assays of the field queenright nests were the same as in the queenless assays. Inside the arena, the beans infested with mealybugs were placed in two acrylic cylinders (5 cm diameter and 1 cm height) to elevate the colony and improve the accuracy of the observations. The non-exposed and the ant-exposed treatments were replicated 62 and 64 times, respectively.

## (d) Effect of previous ant attendance on parasitoid behaviour

For both types of ant colonies, parasitic wasp searching behaviour was recorded in mealybug colonies (i) non-exposed to ants and (ii) ant-exposed. A female wasp was released in an arena with the mealybug colony and the following behaviours were recorded: (i) arrival of the wasp in the mealybug colony (i.e. spent more than 3 s in the colony), (ii) total time spent in the colony, (iii) whether it left the colony (i.e. spent more than 3 s out of the colony), (iv) number of times it left the colony and (v) the total number of times she inserted her ovipositor in a host. Each observation begun 1 min after the parasitic wasp was released in the arena. The observation ended when the wasp did not locate the colony within 30 min or when the wasp stood, rested or walked for more than 5 min without contacting hosts after locating the colony.

To account for potential temporal effects, equal numbers of each treatment were tested each day in both assays, randomizing the order of testing between days. Arenas were used once per day, cleaned with alcohol and left to dry for at least 24 h. All observations were carried out between 10.00 and 15.00 h.

## (e) Effect of ant cue infochemicals on parasitoid behaviour

The effect of *L. grandis* cue extracts on the behaviour of *A. vladimiri* was further investigated with a non-choice bioassay and extracts of the ant cues. Five extra trail extracts (approx. $960 \times 5 = 4800$ ant-equivalents), obtained in the laboratory as described below from queenless ant nests (§2(f)), were gathered and used to treat filter paper squares for the bioassays. Papers treated with ant cues were impregnated with 50 µl of the pentane solution of trail extracts (approx. 320 ant-equivalents). Control papers were impregnated with the same volume of pentane (50 µl). Papers were used for experiments 5 min after they were impregnated to allow the solvent to evaporate. A drop of honey (75%) was provided on treated and control paper squares ($1.5 \times 1.5$ cm). Papers were left in the middle of a glass Petri dish (5 cm diameter), into which a single wasp was then released. Petri dishes were used once per day, cleaned with alcohol and left to dry for at least 24 h.

After allowing the parasitic wasp to settle (1 min), the proportion of wasps that contacted the paper and the time spent on the paper was measured for a period of 10 min. Each treatment was replicated 15 times. To account for potential temporal effects, equal numbers of each treatment were tested each day, randomizing the order of testing between days.

## (f) Composition of ant infochemicals
### (i) Collection of chemical trails

The collection of chemical trails left by *L. grandis* was performed using Teflon-coated wires [30], with slight differences between the queenless ant nests and field ant nests. For the queenless ant nests, metal wires ($25 \times 0.5$ cm diameter) previously washed with ethanol were coated with Teflon tape and were employed as bridges to connect ant nests (total 300 ants) with boxes containing a sucrose solution feeder. To obtain control Teflon-coated bridges, the same procedure was followed but boxes did not have ants.

For the field queenright ant nests, we used a similar methodology as for the behavioural observations. The same boxes were placed on the foraging path of the ants, but bean pods infested with mealybugs were placed inside a smaller plastic box ($16.5 \times 11 \times 6$ cm) coated with Tangle-Trap® (Tanglefoot, Grand Rapids, MI, USA). Metal wires ($25 \times 0.5$ cm diameter) coated with Teflon tape were also employed as bridges to connect the inside of the small plastic boxes with the outside bigger box. Therefore, ants searching in the plastic containers had to walk over the Teflon-coated bridges to reach the mealybug colonies. To obtain control Teflon-coated bridges, the same procedure was followed but the entrance holes to the plastic container were blocked with clay to exclude ants.

In both experiments, ants were allowed to forage and cross the coated bridges for 24 h. Considering 8 h of effective activity of the total 24 h that the bridge was coated and one ant crossing the bridge each 30 s, each trail extract was considered to contain

approximately 960 ant-equivalents. This is likely a conservative estimation as ants remain active during the night. Bridges were carefully removed when ants were not crossing the bridges and Teflon tapes were extracted with 3 ml of pentane (HPLC grade, Sigma-Aldrich, Madrid, Spain) to obtain the cues left behind by the ants, which were subsequently analysed by gas chromatography coupled with mass spectrometry (GC/MS). The control Teflon-coated bridges were extracted in an identical way. Five replicates of chemical trails and four replicates of control samples were collected for the queenless ant nests, while 10 replicates of chemical trails and five replicates of control samples were collected for the field ant nests.

### (ii) Chemical analysis

Chemical trail and control extracts were concentrated to approximately 10 µl under gentle helium flow, and 2 µl were analysed by GC/MS. All injections were performed on a Clarus 600 GC/MS apparatus (Perkin Elmer Inc., Wellesley, PA, USA) equipped with a 30 m × 0.25 × 0.25 fused-silica capillary column (Zebron ZB-5MS, Phenomenex Inc., Torrance, CA, USA). Extracts were injected in splitless mode with the oven programmed at 100°C for 1 min, raised at 10°C min$^{-1}$ up to 180°C, maintained for 1 min and then 5°C min$^{-1}$ up to 280°C with 20-min hold. Injector temperature was set at 250°C and helium at 1 ml min$^{-1}$ was used as carrier gas. The detection was performed in EI mode at 70 eV with ionization source and the transfer line set at 180°C and 250°C, respectively. Scan mode was employed ($m/z$ 35–500), and tentative identification was based on retention indices according to alkane standards and diagnostic ions reported in the literature, as there is no commercial sources for these cuticular hydrocarbons [35–39].

### (g) Statistical analysis

We compared the total time spent in the colony using ANOVA. The normality assumption was assessed using Shapiro's test, and the homoscedasticity assumption was assessed with Levene's test. Individuals that did not find the mealybug colony within the given time frame were excluded from further analysis. The total time spent on the filtered paper was not normally distributed and was analysed with the Wilcoxon test. Proportional and count data were analysed with generalized linear models (GLMs). Initially, we assumed a Poisson error variance for count data (number of times the wasp left the colony, and number of stings per colony) and a binomial error variance for proportional data (colony detection; proportion of colonies with at least one host stung by the parasitic wasp; proportion of filter papers detected by the parasitic wasp). We assessed the assumed error structures by a heterogeneity factor equal to the residual deviance divided by the residual degrees of freedom. If we detected an over- or under-dispersion, we re-evaluated the significance of the explanatory variables using an $F$-test after rescaling the statistical model by a Pearson's $X^2$ divided by the residual degrees of freedom [40]. We present the means of untransformed proportion and count data (in preference to less intuitive statistics such as the back-transformed means of logit-transformed data).

Principal component analysis (PCA) was performed to visualize, through score and loading plots, differences in the chromatographic peak areas of all compounds in the four experimental cases (laboratory assay: queenless ant nests and controls; field assays: field queenright ant nests and controls). The chromatographic peak areas of all compounds were integrated for each replicate. The resulting data were arranged in a matrix of 24 rows (replicates) and 14 columns (chemical compounds as variables). In this dataset, the zero value was assigned to those compounds not detected in a given experimental case. The minimum value of peak area was around $10^4$ units, the median was approximately $10^5$ units and the maximum was approximately $10^6$ units. For compounds detected at trace levels, below the

integration threshold, we used $10^3$ units as an area value, which is 1 log-unit below the minimum integrated area. Then, to normalize the data distribution, area values were transformed by applying the quadratic root transformation. The *prcomp* function was employed to perform the PCA, and the number of principal components to be considered was determined by examining their eigenvalues ($\lambda$) and the proportion of variances by using the *get_eigenvalue* function in the factoextra package. The *ggplot* function in the package ggplot2 was employed to visualize the scores. All data analyses were performed with the R v.3.6.3 statistical package [41].

## 3. Results

### (a) Mealybug colonies exposed to queenless ant nests

Fifty-five per cent of the *A. vladimiri* females located the *P. citri* colonies, and this was independent of the previous exposure of the colonies to *L. grandis* ants ($\chi_1^2 = 0.01$, $p = 0.92$) (figure 1*a*). Once *A. vladimiri* had arrived in the colony, the total time spent foraging in the colony was not affected by ant exposure ($F_{1,60} = 2.17$, $p = 0.15$) (figure 1*b*). However, the proportion of parasitic wasps that left *P. citri* colonies exposed to ants at least once was two times higher than in non-exposed colonies ($\chi_1^2 = 11.25$, $p < 0.001$) (figure 1*c*). Similarly, the number of times that *A. vladimiri* females left the colony was two and half times higher in ant-exposed colonies ($\chi_1^2 = 14.27$, $p = 0.0051$) (figure 1*d*).

The proportion of *A. vladimiri* females that stung at least one mealybug per colony was significantly higher in non-exposed colonies ($0.74 \pm 0.08$) than in colonies exposed to ants ($0.35 \pm 0.09$) ($\chi_1^2 = 75.73$, $p = 0.0019$). Parasitic wasps stung more hosts in non-exposed colonies ($2.8 \pm 0.6$) than in colonies exposed to ants ($1.6 \pm 0.5$) ($F_{1,60} = 5.02$, $p = 0.029$).

### (b) Mealybug colonies exposed to field ant nests

Seventy per cent of the *A. vladimiri* females detected the *P. citri* colonies, and this was independent of the previous exposure of the colonies to *L. grandis* ants ($\chi_1^2 = 152.46$, $p = 0.76$) (figure 2*a*). Once *A. vladimiri* females had arrived in the colony, the total time spent foraging on the colony was not affected by ant exposure ($F_{1,87} = 0.29$, $p = 0.59$) (figure 2*b*). However, the proportion of parasitic wasps that left *P. citri* colonies exposed to ants at least once was two times higher than in non-exposed colonies ($F_{1,87} = 120.88$, $p = 0.023$) (figure 2*c*). Moreover, the number of times that *A. vladimiri* females left the colony was four times higher when the colonies had been exposed to ants ($\chi_1^2 = 78.3$, $p = 0.004$) (figure 2*d*).

All *A. vladimiri* females stung at least one mealybug per colony in the non-exposed colonies, whereas they left 19% of the ant-exposed colonies without stinging any host. Parasitic wasps stung hosts more frequently in non-exposed colonies ($6.5 \pm 0.6$) than in colonies previously exposed to ants ($3.7 \pm 0.4$) ($F_{1,87} = 16.9$, $p < 0.001$).

### (c) Effect of ant infochemicals on parasitic wasp behaviour

The effect of *L. grandis* cue extracts on the behaviour of *A. vladimiri* was further investigated with a non-choice bioassay and filter papers impregnated with or without extracts of the ant cues. More than 80% of the *A. vladimiri* females detected the filter papers, and this was independent of the

*Proc. R. Soc. B* **288**: 20201684

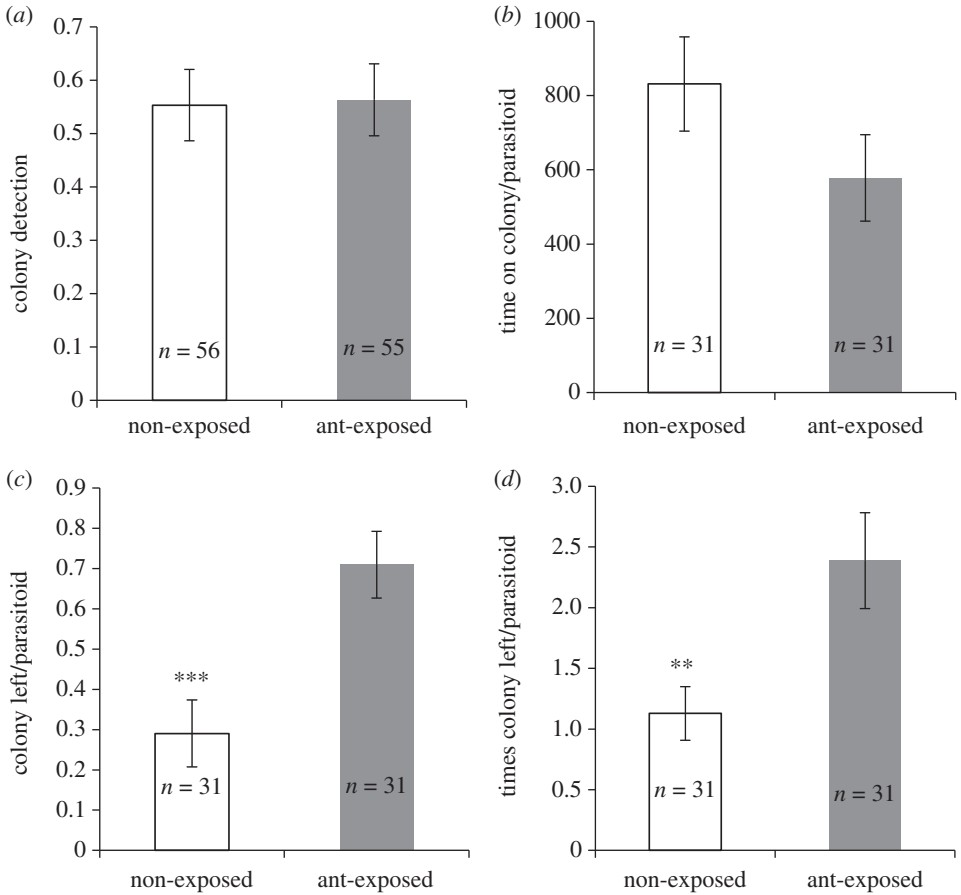

**Figure 1.** Behaviours of the parasitic wasp *A. vladimiri* when it searched in colonies of *P. citri* non-exposed or exposed to *L. grandis* ants from laboratory colonies without queens. (*a*) Proportion of colonies detected. (*b*) Time spent searching in the colony. (*c*) Proportion of colonies left at least once. (*d*) Number of times that the parasitoid left a colony. Columns represent the mean ± s.e., and the number of replicates is presented in each column. Asterisks indicate significant differences between treatments (GLM, *p < 0.05, **p < 0.01, ***p < 0.001).

treatment (control paper: $0.8 \pm 0.1$; paper impregnated with the extract of ant cues: $0.9 \pm 0.1$) ($F_{1,28} = 0.22$, $p = 0.64$). Once *A. vladimiri* females went up to the paper, they spent three times more time on control papers than on papers impregnated with ant cues (control paper: $162.3 \pm 48.4$ s; paper impregnated with the extract of ant cues: $43.6 \pm 13.9$ s) ($W = 122.5$; $p = 0.017$).

## (d) Chemical composition of ant cues

GC/MS analysis showed a series of long-chain saturated hydrocarbons that were consistently present in the pentane extracts of the Teflon-coated bridges (table 1; electronic supplementary material, figure S2). Eight of these hydrocarbons were only detected in the extracts of bridges used by ants during foraging, which were tentatively identified as monomethyl- and dimethyl-branched alkanes, ranging from 28 to 33 carbon atoms. These eight compounds have been previously reported as cuticular compounds of ants (table 1).

The multivariate PCA showed that the chemical profile of the samples differed in composition between those collected in bridges with and without ants (controls) (figure 3). The first two principal components, PC1 and PC2, correspond to the directions with the maximum amount of variation in the dataset, obtaining eigenvalues greater than 1 ($\lambda = 7.08$ and 4.70, respectively) and accounting for 50.6% and 33.6%, respectively, of the total data variability. The compounds responsible for these differences appear grouped on the left side of the plot, corresponding with those only detected in

the extracts of bridges used by ants, regardless if they were from laboratory queenless ant nests (qan) or queenright ant nests (fan).

## 4. Discussion

The parasitic wasp *A. vladimiri* modified its behaviour when it searched in colonies of the mealybug *P. citri* that had been tended by the ant *L. grandis*. In a further bioassay, female wasps responded differentially to filter papers impregnated with extracts of these ant cues than to control papers. Chemical analyses showed that *L. grandis* leaves a complex of cuticular hydrocarbons when it attends mealybug colonies. These cuticular hydrocarbons are also deposited by other ant species [21,37]. *Anagyrus vladimiri* did not discriminate from a distance between a host patch that had been previously attended by ants or a paper impregnated with ant extracts and the respective controls. Therefore, these results suggest that parasitic wasps of honeydew-producing insects can detect the previous presence of mutualistic ants through contact infochemicals.

*Anagyrus vladimiri* wasps modified their foraging behaviour by leaving colonies more frequently and stinging fewer hosts in colonies previously exposed to *L. grandis*. Wasps of the genus *Anagyrus* are attacked and killed by ants when they search in mealybug colonies [7,9,43]. Therefore, this altered foraging behaviour of *A. vladimiri* is likely the result of the detection of a cue that informs the parasitic wasp

Proc. R. Soc. B 288: 20201684

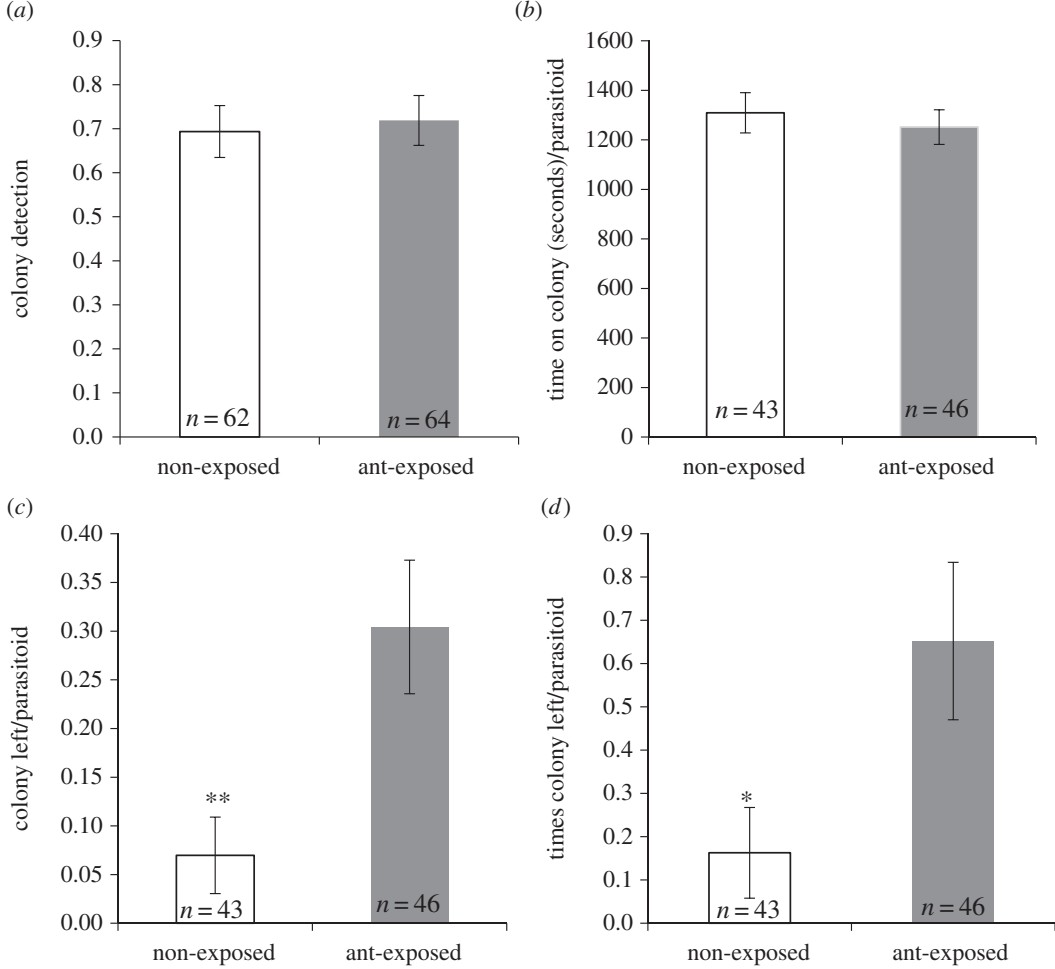

**Figure 2.** Behaviours of the parasitic wasp *A. vladimiri* when it searched in colonies of *P. citri* non-exposed or exposed to *L. grandis* ants from field colonies. (*a*) Proportion of colonies detected by the parasitoid. (*b*) Time spent searching on the colony by the parasitoid. (*c*) Proportion of colonies left at least once by the parasitoid. (*d*) Number of times that the parasitoid left a colony. Columns represent the mean ± s.e., and the number of replicates is presented in each column. Asterisks indicate significant differences between treatments (GLM, *$p < 0.05$, **$p < 0.01$).

that ants are around. Previous studies have demonstrated that parasitic wasps of honeydew producers also alter their behaviour when they are sprayed with extracts of the pygidial gland of *Tapinoma melanocephalum* (Fabricius) (Hymenoptera: Formicidae) ants [44]. Pygidial gland secretions are used by ghost ants, such as *T. melanocephalum*, during aggressive encounters [45,46]. To the best of our knowledge, our study is the first record of parasitic wasps of honeydew-producing hemipterans modifying their behaviour when ants are not physically present in the colony.

The chemical analyses showed that both queenless and queenright colonies of *L. grandis* deposited a complex of cuticular hydrocarbons when attending the mealybugs. We carried out the second experiment with queenright colonies because the absence of queens in ant nests can affect the behaviour of workers. For example, queenless colonies have lower activity levels, change collective behaviour and are more susceptible to diseases [33,47]. These changes might also affect their search for food sources and/or communication by workers. However, our chemical analysis showed that the complex of cuticular hydrocarbons deposited by *L. grandis* workers was qualitatively similar for queenless and queenright nests. This result explains the similar response of *A. vladimiri* to colonies exposed to queenless and queenright nests in our behavioural assays.

Our last bioassay, with filter paper impregnated with ant cue extracts, demonstrates that *A. vladimiri* was able to

detect these cuticular hydrocarbons. Starved wasps spent three times less time on papers impregnated with ant cue extracts than on control papers. The parasitic wasp *A. ervi* also detects the cuticular hydrocarbons deposited by its intra-guild predator *C. septempunctata* and uses them as a cue to avoid predation [17]. In contrast to these results, Appiah *et al.* [48] found that volatile cues, instead of cuticular hydrocarbons, deposited by the African weaver ant *Oecophylla longinoda* (Latreille) (Hymenoptera: Formicidae) alter the preference of the wasp *Fopius arisanus* (Sonan) (Hymenoptera: Braconidae) towards fruits infested by eggs of its tephritid hosts (Diptera: Tephritidae) before landing on them. In that system, both the weaver ant and the wasp are natural enemies of tephritid flies [22,48]. In our study, the percentage of mealy-bug colonies located by *A. vladimiri* was not influenced by the presence of ant infochemicals, suggesting that volatile cues did not mediate it. Beside the cuticular hydrocarbons, *A. vladimiri* might have used other cues that were not measured in our assays. For example, mealybugs might behave differently or change honeydew composition when they are tended by ants and parasitoids might detect these changes. It is known that mealybugs change the composition of their honeydew when they are tended by ants [49]. Further studies are necessary to exclude these potential factors.

Although we did not measure the period during which the *L. grandis* infochemicals affected *A. vladimiri*'s behaviour, it is

**Table 1.** List of compounds detected in pentane extracts of Teflon-coated bridges employed by ants during foraging. Compounds that were detected only in bridges with ants are in bold.

| code | retention time (min) | compound[a] | m.w. | diagnostic EI ions | literature | mean peak areas (n/n)[b] | | | |
| | | | | | | laboratory assay | | field assay | |
| | | | | | | queenless nest | control | queenright nest | control |
| --- | --- | --- | --- | --- | --- | --- | --- | --- | --- |
| c1 | 27.66 | heptacosane | 380 | | | 546 919 (5/5) | 393 766 (4/4) | 119 9 556 (10/10) | 3 461 370 (5/5) |
| c2 | 28.13 | **methylheptacosane**\*\* | 394 | 196/197, 224/225, 253, 281 | [35,36,38,39,42] | **633 471 (5/5)** | 0 | **295 930 (10/10)** | 0 |
| c3 | 28.72 | **3-methylheptacosane** | 394 | 56, 365/366 | [38,39,42] | **137 277 (5/5)** | 0 | **178 182 (10/10)** | 0 |
| c4 | 29.11 | octacosane | 394 | | | 369 049 (5/5) | 273 575 (4/4) | 727 574 (10/10) | 2 555 139 (5/5) |
| c5 | 29.29 | squalene | 410 | | | 500 776 (5/5) | 512 562 (4/4) | 1 001 749 (10/10) | 1 342 106 (5/5) |
| c6 | 30.53 | nonacosane | 408 | | | 329 147 (5/5) | 242 022 (4/4) | 786 718 (10/10) | 2 105 463 (5/5) |
| c7 | 30.99 | **methylnonacosane**\*\* | 422 | 196/197, 224/225, 253 | [36,38,39,42] | **512 367 (5/5)** | 0 | **433 557 (10/10)** | 0 |
| c8 | 31.71 | **3-methylnonacosane** | 422 | 56/57, 392/393 | [36,38,39] | **134 420 (5/5)** | 0 | **33 614 (10/10)** | 0 |
| c9 | 32.14 | triacontane | 422 | | | 247 577 (5/5) | 147 399 (4/4) | 555 910 (10/10) | 1 363 138 (5/5) |
| c10 | 32.67 | **methyltriacontane**\*\* | 436 | 169, 183, 197, 210, 252, 266, 281, 295 | [38] | **63 771 (5/5)** | 0 | **158 442 (9/10)** | 0 |
| c11 | 33.14 | **methyltriacontane**\*\* | 436 | 127, 197 | [35,42] | **90 904 (5/5)** | 0 | **104 246 (7/10)** | 0 |
| c12 | 34.05 | hentriacontane | 436 | | | 222 428 (5/5) | 99 557 (4/4) | 396 615 (10/10) | 911 819 (5/5) |
| c13 | 34.67 | **methylhentriacontane**\*\* | 464 | 196/197, 224/225, 252/253, 281 | [35,36,38,42] | **341 527 (5/5)** | 0 | **764 582 (10/10)** | 0 |
| c14 | 35.22 | **dimethylhentriacontane**\*\* | 464 | 168/169, 196/197, 224/225, 238/239, 266/267, 294/295 | [35,38,42] | **415 111 (5/5)** | 0 | **323 389 (10/10)** | 0 |

[a]Compounds noted with (\*\*) contained a series of positional isomers.
[b](n/n) Number of replicates in which the cuticular compounds were detected and total number of replicates for each experimental case.

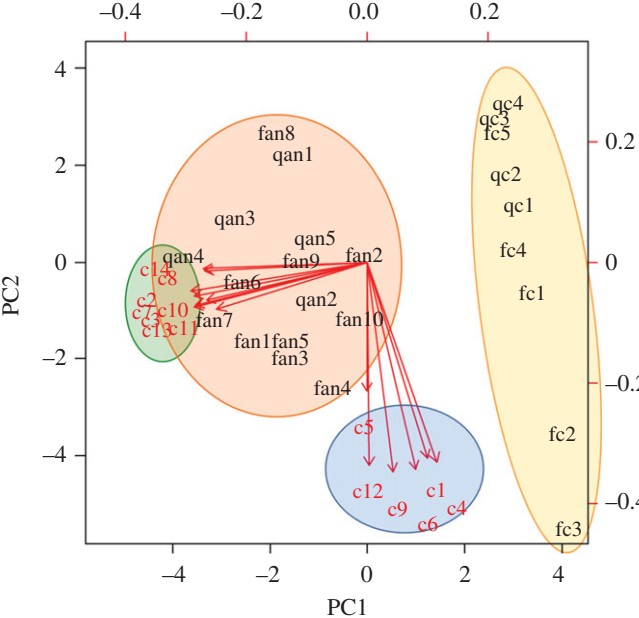

**Figure 3.** PCA of the integrated chromatographic peak areas of the compounds (c1–c14, see table 1 for respective compound names) detected in the extracts of Teflon-coated bridges used by ants (qan: laboratory queenless ant nests; fan: field queenright ant nests) and in extracts of control bridges (fc: field controls and qc: laboratory controls). Prior to the analysis, data (i.e. peak area values) were transformed using the quadratic root and autoscaled. Samples of bridges used by ants and samples of control bridges are grouped inside the orange and yellow ellipses, respectively. The compounds detected in the samples of bridges used by ants and in control bridges are grouped inside the green and blue ellipses, respectively. (Online version in colour.)

known that ant trails can remain detectable in the field for variable periods depending on their identity, function or the foraging strategy. For example, trails can last from several minutes, when used for rapid-response recruitment, to several days and even weeks when foraging on long-lived food sources, such as colonies of honeydew producers [50,51]. If this is the case of *L. grandis* infochemicals, *A. vladimiri* might detect them even several days after the ants abandoned the colony. This is important because ants reduce their activity during some parts of the day and can leave some colonies unattended. For example, Pekas *et al.* [27] demonstrated that *L. grandis* reduce their foraging activity at midday in the summer. During this period, a colony of honeydew producers can remain 'undefended' by ants and parasitic wasps can benefit by being alert to their return.

*Anagyrus vladimiri* exhibits characteristics of an egg-limited species, with a long adult lifespan but a limited number of eggs, which spends more time assessing and choosing a suitable host and laying an egg [3,9]. Therefore, its decision to abandon suitable hosts in the presence of ant cues may maximize its adult lifespan and increase its chances to lay more eggs in the future. However, ant detection and avoidance had also a cost for *A. vladimiri* as females stung and likely parasitized fewer hosts. This trade-off between the acquisition of resources and the avoidance of predation is one of the most prominent trade-offs in ecology and can be also affected by the quality of the visited patch/colony [14].

Overall, the results of the present study suggest that the parasitic wasp *A. vladimiri* alters its foraging decisions in response to the detection of *L. grandis* infochemicals and, as a consequence, it stings fewer hosts and leaves the patch/colony more frequently. Based on the chemical analysis and the behaviour of the parasitic wasp, we suggest that this is

mediated by the cuticular hydrocarbons the ants deposit on the mealybug colony. These novel results are in agreement and complement several studies that report reduced performance of parasitic wasps in the physical presence of antagonistic ants [7,10]. As mutualistic relationships between ants and honeydew producers are ubiquitous, we expect that many parasitic wasp species, sharing similar behavioural and reproductive characteristics with *A. vladimiri*, would exhibit similar responses. Finally, further studies should (i) determine the effect of ant-avoidance by parasitic wasps during its whole adult lifespan (i.e. long-term studies); (ii) determine the effect of previous physical encounters with ants (i.e. experienced versus unexperienced wasps) and (iii) compare the trade-offs of ant-avoidance in egg- versus time-limited wasps.

Data accessibility. The datasets supporting this article can be obtained from the Dryad Digital Repository at: https://doi.org/10.5061/dryad.stqjq2c1n [52].

Authors' contributions. A.T. conceived the ideas. A.M., J.H. and A.T. designed the experiments. A.M. and J.H. collected and analysed the behavioural data. S.V. and V.N.L. collected and analysed the chemical compounds of the extracts. A.M., S.V. and A.T. analysed the data. All authors wrote and approved the final version of the manuscript.

Competing interests. We declare we have no competing interests.

Funding. This research was partially funded by Instituto Nacional de Investigaciones Agrarias Project RTA2017-00095 and the Conselleria d'Agricultura, Pesca i Alimentació de la Generalitat Valenciana. A.M. was a recipient of a MSc scholarship from Bodossaki Foundation (Greece).

Acknowledgements. We thank two anonymous reviewers and editor for their comments and suggestions on the manuscript. We also thank Koppert Biological Systems for providing *Anagyrus* used during the bioassays and the State Insectary of Valencia for providing *P. citri*. We acknowledge D. Sánchez-García, A. Micó, M. Calvo-Agudo, P. Bru and J. Catalán for their valuable help during the experiments.

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
