## [Reviewer comments · Proceedings of the Royal Society B: Biological Sciences]

Review History

RSPB-2020-1684.R0 (Original submission)

Review form: Reviewer 1

Recommendation

Accept with minor revision (please list in comments)

Scientific importance: Is the manuscript an original and important contribution to its field?

Excellent

General interest: Is the paper of sufficient general interest?

Excellent

Quality of the paper: Is the overall quality of the paper suitable?

Good

Is the length of the paper justified?

Yes

Should the paper be seen by a specialist statistical reviewer?

No

Do you have any concerns about statistical analyses in this paper? If so, please specify them explicitly in your report.

Yes

It is a condition of publication that authors make their supporting data, code and materials available - either as supplementary material or hosted in an external repository. Please rate, if applicable, the supporting data on the following criteria.

Is it accessible?

Yes

Is it clear?

Yes

Is it adequate?

Yes

Do you have any ethical concerns with this paper?

No

Comments to the Author

The minor comments I suggest to review are:

L 46: Should be "cuticle" instead "cuti"

L 80-81: For your second objective "evaluate the cost, measured as the number of parasitization per colony, that results from behavioural response to ant cues" I suggest that maybe you can eliminate "the number of parasitization per colony" since you did not count parasitized hosts, at least there is not evidence through the document of this evaluation.

L 86: Riso in parenthesis as the authors did in previous names.

L 102: Reference with number (34) instead (2004).

L 113: Delete "h", photoperiods are normally in hours and it is not needed.

L 107: What do the authors mean with "natural daylight" if the bioassays were in lab conditions?

L 133: What was the reason of 48 hours of starving ants?, why not 24?

L 182: How did the authors know the ovipositor was inserted in a host? These species are very small and sometimes is very difficult to determine this behaviour.

L 184: Eliminate "minimum" unless you have a maximum time. If not, what was the maximum time of the observations?

L 186: How many replicates were done for this bioassay?

L 198: The authors put a drop of honey in the paper, do you think it could affect the results? Do the authors think having a third treatment with only honey could be helpful to discard some effect?

L 282: The authors report ANOVA results but you said (L 249) it should be a Wilcoxon test since distribution were not normal.

L 295: Same comment as before.

L 319: Which references reported cuticular compounds of ants?

L 345: Add order and family of *T. melanocephalum*.

L 355: *L. niger* is first mentioned in the document, you should write the full scientific name and add the discoverer, order and family like with the previous species in the document.

L 365: Add discoverer's name.

L 501: It is Angeles instead "Angelos"

Figure 1: In methodology the authors mention that there were 55 replicates for this bioassay but in the graphics you show it is 55 replicates for ant-treatment and 56 for non-exposed, what happened here? and what about the n=31?

Figure 2: Same comment as before, you mention 63 replicates in the methodology and here in graphics you show 62 for control treatment and 64 for the ant-treatment. and n=43 and n=46 in the following graphics.

Review form: Reviewer 2

Recommendation

Major revision is needed (please make suggestions in comments)

Scientific importance: Is the manuscript an original and important contribution to its field?

Good

General interest: Is the paper of sufficient general interest?

Good

Quality of the paper: Is the overall quality of the paper suitable?

Excellent

Is the length of the paper justified?

Yes

Should the paper be seen by a specialist statistical reviewer?

No

Do you have any concerns about statistical analyses in this paper? If so, please specify them explicitly in your report.

No

It is a condition of publication that authors make their supporting data, code and materials available - either as supplementary material or hosted in an external repository. Please rate, if applicable, the supporting data on the following criteria.

Is it accessible?

Yes

Is it clear?

Yes

Is it adequate?

Yes

Do you have any ethical concerns with this paper?

No

Comments to the Author

In the study at hand, the authors compare behavioural interactions between a parasitoid and a phloem feeder in relation to previous ant exposure. This is done in laboratory assays by confronting parasitoids with ant tended or control citrus mealybugs as well as with chemotactile cues of ants. In addition, the authors analysed the chemical composition of ant trails to identify potential substances involved in ant detection.

In line with their expectations, a higher proportion of the parasitoids left ant-exposed colonies and they left the ant-exposed colonies more often compared to control colonies - with consequences for oviposition opportunities. There were no differences between experiments with or without queens in ant colonies. Furthermore, parasitoids spent three times more time on control papers than on papers treated with ant cues. The results suggest, that parasitoids are able to recognize ant chemical cues upon contact and avoid such colonies in order to reduce intraguild

interference with ants.

The research questions addressed by the authors is interesting and important for basic and applied ecology (e.g. pest control). In my opinion, the authors applied an impeccable experimental design with high sample sizes to test their hypotheses and the results are very convincing to me. The manuscript is well structured and written. However, a couple of sentences are somewhat unclear or too farfetched. Below, I provide some suggestions that might help to further improve the manuscript.

Detailed comments:

Introduction

L 53: please reformulate the sentence "...and exuviae and by transporting them" what is 'them' so far the sentence is about honeydew-producing colonies.

L 58-61: In my opinion this statement is too strong: Parasitoids are highly mobile and can easily escape ants by flying away. Soft skinned, immobile aphid predators such as syrphid or ladybeetle larva are at much greater risk.

L 73-76: In the introduction, I miss the reasoning why the presence of a queen in the ant colony should affect the response of parasitoids. The two experiments (without queen and queenright colonies) come as a surprise. This should be introduced or at least be explained in the method section.

L 77: Sentence is too long and unclear: "can detect" unclear what they can detect? Please reformulate.

L 79: Please formulate your hypotheses more directly to what actually is quantified in the experiments: detection is not quantified here (e.g. as an electrophysiological signal) - it is about behavioural changes: which changes are expected for the respective behavioural elements?

L 80-81: the same thing as above: no costs are quantified instead oviposition attempts are counted - it is unclear if this leads to costs for parasitoids (it might be that it is advantageous to lay the eggs into ant tended colonies because ants kill and deter Hemiptera predators or other parasitoids - maybe an additional aspect for the discussion).

L 85: In my opinion the description of the study system should either be before the hypotheses or in the beginning of the method section.

M & m's

L 127: Fluon® is a brand like Parafilm®, so it should be mentioned as such.

L 145: ((

L 192-194: "Five extra trail extracts [...] were gathered" How?

L 203-204: "Each treatment was repeated 15 times." Are these repeated measurements or independent replicates?

L 256-258: "If we detected and over- or underdispersion, we re-evaluated the significance of the explanatory variables using a F test after rescaling the statistical model by a Person's Chi-square divided by the residual degrees of freedom." This sounds like a method from medieval times - why not simply using glm with quasi-binomial/quasi-Poisson error distribution?

Discussion

L 350: Here, I would have expected a joint discussion of queenless and queenright colonies including chemical cues and similarities in behavioural responses. So far it remains unclear why to compare queenless/queenright colonies and why to expect differences (see comment above).

Decision letter (RSPB-2020-1684.R0)

09-Oct-2020

Dear Dr Tena:

Your manuscript has now been peer reviewed and the reviews have been assessed by an Associate Editor. The reviewers' comments (not including confidential comments to the Editor) and the comments from the Associate Editor are included at the end of this email for your reference. As you will see, the reviewers and the Editors have raised some concerns with your manuscript and we would like to invite you to revise your manuscript to address them.

Research ethics:

Use of animals and field studies:

It is a condition of publication that you make available the data and research materials supporting the results in the article. Please see our Data Sharing Policies (<https://royalsociety.org/journals/authors/author-guidelines/#data>). Datasets should be deposited in an appropriate publicly available repository and details of the associated accession number, link or DOI to the datasets must be included in the Data Accessibility section of the article (<https://royalsociety.org/journals/ethics-policies/data-sharing-mining/>). Reference(s) to datasets should also be included in the reference list of the article with DOIs (where available).

Please submit a copy of your revised paper within three weeks. If we do not hear from you within this time your manuscript will be rejected. If you are unable to meet this deadline please let us know as soon as possible, as we may be able to grant a short extension.

Best wishes,
Dr Sasha Dall
mailto: proceedingsb@royalsociety.org

Associate Editor
Comments to Author:

The paper was evaluated by two reviewers, who both had overall favorable opinions of the work. Referee 1 asks for some minor revisions, whereas Referee 2 is asking for some statistical re-analysis and more information/interpretation, especially about the relevance of comparing queenless vs queenright colonies. I recommend the authors revise the manuscript along with all of the suggestions made by the reviewers.

Reviewer(s)' Comments to Author:

Referee: 1

Comments to the Author(s)

The minor comments I suggest to review are:

L 46: Should be "cuticle" instead "cuti"

L 80-81: For your second objective "evaluate the cost, measured as the number of parasitization per colony, that results from behavioural response to ant cues" I suggest that maybe you can eliminate "the number of parasitization per colony" since you did not count parasitized hosts, at least there is not evidence through the document of this evaluation.

L 86: Risso in parenthesis as the authors did in previous names.

L 102: Reference with number (34) instead (2004).

L 113: Delete "h", photoperiods are normally in hours and it is not needed.

L 107: What do the authors mean with "natural daylight" if the bioassays were in lab conditions?

L 133: What was the reason of 48 hours of starving ants?, why not 24?

L 182: How did the authors know the ovipositor was inserted in a host? These species are very small and sometimes is very difficult to determine this behaviour.

L 184: Eliminate "minimum" unless you have a maximum time. If not, what was the maximum time of the observations?

L 186: How many replicates were done for this bioassay?

L 198: The authors put a drop of honey in the paper, do you think it could affect the results? Do the authors think having a third treatment with only honey could be helpful to discard some effect?

L 282: The authors report ANOVA results but you said (L 249) it should be a Wilcoxon test since distribution were not normal.

L 295: Same comment as before.

L 319: Which references reported cuticular compounds of ants?

L 345: Add order and family of *T. melanocephalum*.

L 355: *L. niger* is first mentioned in the document, you should write the full scientific name and add the discoverer, order and family like with the previous species in the document.

L 365: Add discoverer`s name.

L 501: It is Angeles instead "Angelos"

Figure 1: In methodology the authors mention that there were 55 replicates for this bioassay but in the graphics you show it is 55 replicates for ant-treatment and 56 for non-exposed, what happened here? and what about the n=31?

Figure 2: Same comment as before, you mention 63 replicates in the methodology and here in graphics you show 62 for control treatment and 64 for the ant-treatment. and n=43 and n=46 in the following graphics.

Referee: 2

Comments to the Author(s)

In the study at hand, the authors compare behavioural interactions between a parasitoid and a phloem feeder in relation to previous ant exposure. This is done in laboratory assays by confronting parasitoids with ant tended or control citrus mealybugs as well as with chemotactile cues of ants. In addition, the authors analysed the chemical composition of ant trails to identify potential substances involved in ant detection.

In line with their expectations, a higher proportion of the parasitoids left ant-exposed colonies and they left the ant-exposed colonies more often compared to control colonies – with consequences for oviposition opportunities. There were no differences between experiments with or without queens in ant colonies. Furthermore, parasitoids spent three times more time on control papers than on papers treated with ant cues. The results suggest, that parasitoids are able to recognize ant chemical cues upon contact and avoid such colonies in order to reduce intraguild interference with ants.

The research questions addressed by the authors is interesting and important for basic and applied ecology (e.g. pest control). In my opinion, the authors applied an impeccable experimental design with high sample sizes to test their hypotheses and the results are very convincing to me. The manuscript is well structured and written. However, a couple of sentences are somewhat unclear or too farfetched. Below, I provide some suggestions that might help to further improve the manuscript.

Detailed comments:

Introduction

L 53: please reformulate the sentence "...and exuviae and by transporting them" what is 'them' so far the sentence is about honeydew-producing colonies.

L 58-61: In my opinion this statement is too strong: Parasitoids are highly mobile and can easily escape ants by flying away. Soft skinned, immobile aphid predators such as syrphid or ladybeetle larva are at much greater risk.

L 73-76: In the introduction, I miss the reasoning why the presence of a queen in the ant colony should affect the response of parasitoids. The two experiments (without queen and queenright colonies) come as a surprise. This should be introduced or at least be explained in the method section.

L 77: Sentence is too long and unclear: "can detect" unclear what they can detect? Please reformulate.

L 79: Please formulate your hypotheses more directly to what actually is quantified in the experiments: detection is not quantified here (e.g. as an electrophysiological signal) - it is about behavioural changes: which changes are expected for the respective behavioural elements?

L 80-81: the same thing as above: no costs are quantified instead oviposition attempts are counted - it is unclear if this leads to costs for parasitoids (it might be that it is advantageous to lay the eggs into ant tended colonies because ants kill and deter Hemiptera predators or other parasitoids - maybe an additional aspect for the discussion).

L 85: In my opinion the description of the study system should either be before the hypotheses or in the beginning of the method section.

M & m's

L 127: Fluon® is a brand like Parafilm®, so it should be mentioned as such.

L 145: ((

L 192-194: "Five extra trail extracts [...] were gathered" How?

L 203-204: "Each treatment was repeated 15 times." Are these repeated measurements or independent replicates?

L 256-258: "If we detected and over- or underdispersion, we re-evaluated the significance of the explanatory variables using a F test after rescaling the statistical model by a Person's Chi-square divided by the residual degrees of freedom." This sounds like a method from medieval times - why not simply using glm with quasi-binomial/quasi-Poisson error distribution?

Discussion

L 350: Here, I would have expected a joint discussion of queenless and queenright colonies including chemical cues and similarities in behavioural responses. So far it remains unclear why to compare queenless/queenright colonies and why to expect differences (see comment above).

Author's Response to Decision Letter for (RSPB-2020-1684.R0)

See Appendix A.

Decision letter (RSPB-2020-1684.R1)

13-Nov-2020

Dear Dr Tena:

Your manuscript has now been peer reviewed and the reviews have been assessed by an Associate Editor. The reviewers' comments (not including confidential comments to the Editor) and the comments from the Associate Editor are included at the end of this email for your reference. As you will see, the reviewers and the Editors have raised some concerns with your manuscript and we would like to invite you to revise your manuscript to address them.

We urge you to make every effort to fully address all of the comments at this stage. If deemed necessary by the Associate Editor, your manuscript will be sent back to one or more of the original reviewers for assessment. If the original reviewers are not available we may invite new reviewers. Please note that we cannot guarantee eventual acceptance of your manuscript at this stage.

Research ethics:

Use of animals and field studies:

It is a condition of publication that you make available the data and research materials supporting the results in the article (<https://royalsociety.org/journals/authors/author-guidelines/#data>). Datasets should be deposited in an appropriate publicly available repository and details of the associated accession number, link or DOI to the datasets must be included in the Data Accessibility section of the article (<https://royalsociety.org/journals/ethics-policies/data-sharing-mining/>). Reference(s) to datasets should also be included in the reference list of the article with DOIs (where available).

Please submit a copy of your revised paper within three weeks. If we do not hear from you within this time your manuscript will be rejected. If you are unable to meet this deadline please let us know as soon as possible, as we may be able to grant a short extension.

Best wishes,
Dr Sasha Dall
Editor, Proceedings B
mailto: proceedingsb@royalsociety.org

Associate Editor

Comments to Author:

This manuscript needs additional revision; I would like to give the authors another opportunity to more thoroughly respond to the referee's comments.

First, I suggest changing the title. As is, it is awkwardly worded. I suggest something along the lines of "Parasitic wasps avoid ant-protected aphid hosts via detection of ant chemical signatures" or "Host choice by parasitic wasps in the context of an ant-aphid mutualism is guided by avoidance of ant chemical cues"

Second, in responses to referees, the authors repeatedly used length restrictions as an argument for not responding to several of the queries made by the referees. The authors need to provide more complete responses to several of the referees' comments through changes to the text, and shorten other parts of the manuscript accordingly to meet length guidelines. Thus, another revision is necessary in which the authors more completely respond to ALL referee queries.

Author's Response to Decision Letter for (RSPB-2020-1684.R1)

See Appendix B.

Decision letter (RSPB-2020-1684.R2)

07-Dec-2020

Dear Dr Tena

I am pleased to inform you that your manuscript entitled "Parasitic wasps avoid ant-protected hemipteran hosts via detection of ant cuticular hydrocarbons" has been accepted for publication in Proceedings B.

Open Access

Paper charges

Sincerely,

Dr Sasha Dall

Associate Editor:

Board Member

Comments to Author:

Thank you for your careful attention to the additional comments. The new title is much improved, and I appreciate the more thorough responses to the referee comments.

Appendix A

Moncada, Spain, 23 October 2020

Dear Editor,

Many thanks for taking the time to evaluate our manuscript “**Ant-hemipteran mutualism: Parasitic wasps use cuticular hydrocarbons of ants to avoid them**”, authored by Angelos Mouratidis, Sandra Vacas, Julieta Herrero, Vicente Navarro-Llopis, Marcel Dicke and Alejandro Tena that we submitted for consideration of publication in *Proceedings of the Royal Society B: Biological Sciences*. We are also grateful for the time spent by the two Reviewers and for their valuable comments on the manuscript.

We have carefully considered all comments by the Reviewers and have used them to further improve the manuscript. We hope that the manuscript now meets with the standard of publication in *Proceedings of the Royal Society B: Biological Sciences*.

Looking forward to your response.

Kind regards,

Angelos Mouratidis and Alejandro Tena on behalf of all authors

Reviewer #1:

- 1) **L 46: Should be "cuticle" instead "cuti"**
DONE. Replaced with “cuticular hydrocarbons”.

- 2) **L 80-81: For your second objective "evaluate the cost, measured as the number of parasitization per colony, that results from behavioural response to ant cues" I suggest that maybe you can eliminate "the number of parasitization per colony" since you did not count parasitized hosts, at least there is not evidence through the document of this evaluation.**
DONE. Thanks. As we explained in the Abstract, M&M and Results, we counted the number of hosts stung. We have corrected it: “ii) evaluate the number of hosts stung per colony, that results from the behavioural response to ant cues”.

- 3) **L 86: Risso in parenthesis as the authors did in previous names.**
DONE. Thanks.

- 4) **L 102: Reference with number (34) instead (2004).**
DONE. Thanks.

- 5) **L 113: Delete "h", photoperiods are normally in hours and it is not needed.**
DONE. Thanks.

- 6) **L 107: What do the authors mean with "natural daylight" if the bioassays were in lab conditions?**
DONE. We have removed natural daylight.

- 7) **L 133: What was the reason of 48 hours of starving ants?, why not 24?**
RESPONSE: This was done to ensure high activity of foraging ants in mealybug colonies.

- 8) **L 182: How did the authors know the ovipositor was inserted in a host? These species are very small and sometimes is very difficult to determine this behaviour.**
RESPONSE: *Anagyrus vladimiri* females have a characteristic host handling behaviour. Foraging females continuously tap with their antennae on the plant substrate and when they encounter a host this behaviour stops. Female wasps then perform a 180° turn, extend their ovipositor and insert it in the host. In some cases, females just touched the host without inserting the ovipositor. This behaviour, however, only lasts a few seconds.

- 9) **L 184: Eliminate "minimmum" unless you have a maximum time. If not, what was the maximum time of the observations?**

DONE. We have changed to: “The observation ended when the wasp did not locate the colony within 30 min or when the wasp rested or walked for more than 5 min without contacting hosts after locating the colony”.

L 186: How many replicates were done for this bioassay?

DONE: We agree that the number of replicates was confusing. We have modified the text in lines L 153 for queenless bioassay (“The non-exposed and the ant-exposed treatments were replicated 56 and 55 times, respectively.”) and L 181 for the queenright (“The non-exposed and the ant-exposed treatments were replicated 62 and 64 times, respectively.”)

10) L 198: The authors put a drop of honey in the paper, do you think it could affect the results? Do the authors think having a third treatment with only honey could be helpful to discard some effect?

RESPONSE: We used honey to attract the wasp in both treatments (control paper vs impregnated paper). We don't understand the concern of this methodology.

11) L 282: The authors report ANOVA results but you said (L 249) it should be a Wilcoxon test since distribution were not normal.

RESPONSE: Please, see that the Wilcoxon test was performed for the “total time a parasitoid spent on the filter paper” in section (c) of the results. For the queenless and queenright assays, “the total time spent on the colony” was analyzed with ANOVA (as explained in L 248-249 of the highlighted version).

12) L 295: Same comment as before.

RESPONSE: Please, see our answer above.

13) L 319: Which references reported cuticular compounds of ants?

RESPONSE: The references are found in Table 1, under the column Literature. Several sources are listed for each compound found.

14) L 345: Add order and family of T. melanocephalum.

DONE. Thanks

15) L 355: L. niger is first mentioned in the document, you should write the full scientific name and add the discoverer, order and family like with the previous species in the document.

DONE. Thanks.

16) L 365: Add discoverer's name.

DONE. Thanks.

17) L 501: It is Angeles instead "Angelos"

DONE. Thanks.

18) Figure 1: In methodology the authors mention that there were 55 replicates for this bioassay but in the graphics you show it is 55 replicates for ant-treatment and 56 for non-exposed, what happened here? and what about the n=31?

DONE. Thanks. We have modified the number of replicates in the M&M section as explained above.

Regarding the number of replicates in the colony, the individuals that did not find the colony were excluded from further analyses. Thus, this resulted in 31 parasitoids in each treatment for the queenless experiment. To make it clearer, we have added this explanation in the M&M section “Individuals that did not find the mealybug colony within the given timeframe were excluded from further analysis.” Line 251.

19) Figure 2: Same comment as before, you mention 63 replicates in the methodology and here in graphics you show 62 for control treatment and 64 for the ant-treatment. and n=43 and n=46 in the following graphics.

DONE. The explanation is the same as above for similar comments.

Many thanks for the detailed and constructive review of our manuscript!

Reviewer #2:

In the study at hand, the authors compare behavioural interactions between a parasitoid and a phloem feeder in relation to previous ant exposure. This is done in laboratory assays by confronting parasitoids with ant tended or control citrus mealybugs as well as with chemotactile cues of ants. In addition, the authors analysed the chemical composition of ant trails to identify potential substances involved in ant detection.

In line with their expectations, a higher proportion of the parasitoids left ant-exposed colonies and they left the ant-exposed colonies more often compared to control colonies – with consequences for oviposition opportunities. There were no differences between experiments with or without queens in ant colonies. Furthermore, parasitoids spent three times more time on control papers than on papers treated with ant cues. The results suggest, that parasitoids are able to recognize ant chemical cues upon contact and avoid such colonies in order to reduce intraguild interference with ants.

The research questions addressed by the authors is interesting and important for basic and applied ecology (e.g. pest control). In my opinion, the authors applied an impeccable experimental design with high sample sizes to test their hypotheses and the results are very convincing to me. The manuscript is well structured and written. However, a couple of sentences are somewhat unclear or too farfetched. Below, I provide some suggestions that might help to further improve the manuscript.

Many thanks for your nice words and detailed review.

1) **L 53: please reformulate the sentence “...and exuviae and by transporting them” what is ‘them’ so far the sentence is about honeydew-producing colonies.**
DONE. “them” has been replaced by “honeydew producers”.

2) **L 58-61: In my opinion this statement is too strong: Parasitoids are highly mobile and can easily escape ants by flying away. Soft skinned, immobile aphid predators such as syrphid or ladybeetle larva are at much greater risk.**
DONE: We agree that certain parasitoids may not be vulnerable to ant predation. This for example applies to parasitoids that exhibit rapid oviposition, such as *Coccinnoxenoides permitutus*, while species of the genus *Anagyrus* that need more time to oviposit are more vulnerable (Sime and Daane, 2014). Thus, we have rephrased the sentence: “Parasitic wasps can be vulnerable to these attacks because they remain defenceless when they parasitize honeydew producers or feed on host haemolymph, i.e. host feeding”

3) **L 73-76: In the introduction, I miss the reasoning why the presence of a queen in the ant colony should affect the response of parasitoids. The two experiments (without queen and queenright colonies) come as a surprise. This should be introduced or at least be explained in the method section.**

DONE: We agree that was not very clear in the manuscript. We had omitted this justification due to the limited space of the journal.

We have now added the reasoning to repeat the experiment in the field using queenright colonies (L 157-159): “To corroborate the results obtained with queenless ant nests, we carried out a similar experiment but using field queenright ant nests. The presence of queens in ant nests affects the behaviour and chemical cues used for communication by workers in some ant species [36].”

4) **L 77: Sentence is too long and unclear: “can detect” unclear what they can detect? Please reformulate.**

DONE: This sentence has been shortened and reformulated.

5) **L 79: Please formulate your hypotheses more directly to what actually is quantified in the experiments: detection is not quantified here (e.g. as an electrophysiological signal) – it is about behavioural changes: which changes are expected for the respective behavioural elements?**

RESPONSE. We think that reformulating our hypotheses would make the manuscript longer and we would lose our main hypothesis. Please, see that we have used several behaviours (colony detection, time spend in the colony, proportion colonies left, and number of times the colony were left) and assays (the queenright and queenless bioassays and the ant cue bioassay) to test our main hypothesis: “determine whether parasitic wasps that parasitize honeydew producers can detect the previous presence of mutualistic ants in the colony of the honeydew producers”. In other words, we have used several assays, that are described in the Material and Methods section, to test our

first hypothesis. Moreover, and following the recommendations of this reviewer, we have explained why we carried out the field experiment with queenright colonies in the Material and Methods section.

- 6) **L 80-81: the same thing as above: no costs are quantified instead oviposition attempts are counted – it is unclear if this leads to costs for parasitoids (it might be that it is advantageous to lay the eggs into ant tended colonies because ants kill and deter Hemiptera predators or other parasitoids – maybe an additional aspect for the discussion).**

DONE. We have modified it.

- 7) **L 85: In my opinion the description of the study system should either be before the hypotheses or in the beginning of the method section.**

RESPONSE. We prefer to keep the description of the study system as it is now. In our opinion, it is not part of the material and methods and it would break the reading before the hypotheses.

- 8) **L 127: Fluon® is a brand like Parafilm®, so it should be mentioned as such.**

DONE. Thanks. Also done in L 136.

- 9) **L 145: ((**

DONE. Thanks.

- 10) **L 192-194: “Five extra trail extracts [...] were gathered” How?**

DONE. Thanks. It was an error. We wrote “below” instead of “above”. We have also indicated the section where it is explained: “Five extra trail extracts (ca. $960 \times 5 = 4,800$ ant-equivalents), obtained in the laboratory as described below from queenless ant nests (section f), were gathered and used to treat filter paper squares for the bioassays.”

- 11) **L 203-204: “Each treatment was repeated 15 times.” Are these repeated measurements or independent replicates?**

DONE. Thanks. These were independent replicates. We have changed “repeated” by “replicated”.

- 12) **L 256-258: “If we detected and over- or underdispersion, we re-evaluated the significance of the explanatory variables using a F test after rescaling the statistical model by a Person’s Chi-square divided by the residual degrees of freedom.” This sounds like a method from medieval times – why not simply using glm with quasi-binomial/quasi-Poisson error distribution?**

RESPONSE: Indeed that is exactly what we have done. When the model we fitted showed over- or under-dispersion, we estimated a quasi-dispersion parameter and corrected the model. This quasi-dispersion parameter (σ^2 for binomial and ϕ for Poisson distributions) is estimated with the formula $X^2/(n-p)$;

where X^2 is Pearson's goodness-of-fit statistic, n is the number of samples and p is the number of parameters. Effectively, $n-p$ equals the degrees of freedom in the model. Thus, the part the reviewer quotes from our text here is the definition of a quasi-likelihood model, as defined by McCullagh and Nelder (1989) (equation 4.23, p. 127, also Chapter 9 on Quasi-likelihood functions p. 328).

In practice, when in the models we fitted over- or under-dispersion was found, this was corrected by changing the family of the model to quasipoisson or quasibinomial for count and binary data, respectively.

13) L 350: Here, I would have expected a joint discussion of queenless and queenright colonies including chemical cues and similarities in behavioural responses. So far it remains unclear why to compare queenless/queenright colonies and why to expect differences (see comment above).

DONE: We agree that this was not very clear in the manuscript. Apart from the section added in the M&M, another statement is added in L 360-362 to point out the similarities between the chemical profiles between queenless and queenright nests. We would like to extend this discussion, but we have run out of space according to the journal guidelines.

References

Sime KR, Daane KM. 2014 A Comparison of Two Parasitoids (Hymenoptera: Encyrtidae) of the Vine Mealybug: Rapid, Non-Discriminatory Oviposition Is Favored When Ants Tend the Host. *Environ. Entomol.* **43**, 995–1002. (doi:10.1603/EN13192)

McCullagh, P. and J.A. Nelder, 1989. *Generalized Linear Models*. Chapman and Hall, London

Appendix B

Moncada, Spain, 27 November 2020

Dear Editor,

Many thanks for your time and suggestions. Following your recommendation, we have changed the title to “**Parasitic wasps avoid ant-protected hemipteran hosts via detection of ant cuticular hydrocarbons**”. We really appreciate this suggestion, the new title is more direct and clear.

Regarding the use of length restrictions as an argument for not responding to several of the queries made by the referees, we have found two comments of Reviewer 2 where we mentioned the limitation of space. These are comments 5) and 13) in our “Response to referees” letter. We would like to highlight that, in both cases, we answered to the referee’s comments and explain either why we preferred to keep the manuscript as it was (comment 5) or the modifications that we carried out despite the space limitation (comment 13). In other words, we took into consideration all the recommendations of this reviewer, whose comments have really improved the manuscript. In the new version of the manuscript, we have included these two recommendations and removed some sections as explained in detail below.

5) L 79: Please formulate your hypotheses more directly to what actually is quantified in the experiments: detection is not quantified here (e.g. as an electrophysiological signal) – it is about behavioural changes: which changes are expected for the respective behavioural elements?

DONE. Following the recommendation of the Reviewer and Associate Editor, we have changed to:

“Here, we first: i) determined whether parasitic wasps that parasitize honeydew producers can detect the previous presence of mutualistic ants in the colony of the honeydew producers. For this, we measured changes in colony detection, time spent foraging in the colony, as well as, proportion and number of times that parasitic wasps left the colonies when they searched in colonies non-exposed and previously exposed to ants. We expect that if parasitic wasps detect the previous presence of ants in colonies using ant

cues, they will locate colonies equally, spend less time in colonies previously exposed to ants and leave these colonies more frequently. Then, we ii) evaluated the number of hosts stung per colony. We expect that if parasitic wasps can detect the previous presence of ants in the colony, they will sting more hosts in non-exposed colonies than in colonies exposed to ants. Finally, we iii) identified the chemical cues involved in ant detection. We have addressed these aims through behavioural and chemical assays using laboratory (queenless) and field (queenright) ant nests.”

13) L 350: Here, I would have expected a joint discussion of queenless and queenright colonies including chemical cues and similarities in behavioural responses. So far it remains unclear why to compare queenless/queenright colonies and why to expect differences (see comment above).

DONE. Regarding the second part of the comment, we have modified the M&M section:

“To corroborate the results obtained with queenless ant nests, we carried out a similar experiment but using field queenright ant nests. This is because the presence of queens in ant nests affects the behaviour of workers in some ant species and, therefore, might also change the chemical cues left by workers [32].”

Regarding the first part of the comment, we have modified this paragraph following the recommendation of the reviewer:

“The chemical analyses showed that both queenless and queenright colonies of *L. grandis* deposited a complex of cuticular hydrocarbons when attending the mealybugs. We carried out the second experiment with queenright colonies because the absence of queens in ant nests can affect the behaviour of workers. For example, queenless colonies have lower activity levels, change collective behavior and are more susceptible to diseases [32,46]. These changes might also affect their search for food sources and/or communication by workers. However, our chemical analysis showed that the complex of cuticular hydrocarbons deposited by *L. grandis* workers was qualitatively similar for queenless and queenright nests. This result explains the similar response of *A. vladimiri* to colonies exposed to queenless and queenright nests in our behavioural assays.”

Due to the length restrictions and the inclusion of the new paragraph and detailed explanation of the hypotheses, we have removed i) references from statements supported by more than two references and ii) one paragraph from the discussion:

“The chemical analyses showed that both queenless and queenright colonies of *L. grandis* deposited a complex of cuticular hydrocarbons when attending the mealybugs. These chemical profiles were similar for queenless and queenright nests, in agreement with previous studies on species of the genus *Lasius* [41]. Hydrocarbons are found in the postpharyngeal gland and cuticle of many ant species and serve as species, sex and nestmate recognition cues [49]. In the genus *Lasius*, *L. japonicus* Santschi and *L. nipponensis* Forel (Hymenoptera: Formicidae) are known to have high similarities between their footprint hydrocarbons and cuticular hydrocarbons [50,51]. Furthermore, for the ant *Lasius niger* (L.) (Hymenoptera: Formicidae), the footprint hydrocarbons are known to be deposited passively while they are foraging [41] and actively to deter the dispersal of mutualistic aphids [23].”

We hope that the manuscript now meets with the standard of publication in *Proceedings of the Royal Society B: Biological Sciences*.

Kind regards,

Angelos Mouratidis and Alejandro Tena on behalf all the coauthors